# Creating a foundation for origin of life outreach: How scientists relate to their field, the public, and religion

**Karl Wienand**[1,2]*, **Lorenz Kampschulte**[2], **Wolfgang M. Heckl**[1,2]

**1** Oskar-von-Miller Chair for Science Communication, TUM School of Social Sciences and Technology, Technical University of Munich, Munich, Germany, **2** Deutsches Museum, Munich, Germany

* karl.wienand@tum.de

**Data Availability Statement:** All relevant data are within the paper.

**Funding:** Karl Wienand was funded by the Deutsche Forschungsgemeinschaft (DFG, German Research Foundation) with Project-ID 364653263

## Abstract

Origins of life research is particularly challenging to communicate because of the tension between its many disciplines and its nearness to traditionally philosophical or religious questions. To authentically represent scientists' perspective in a museum exhibition, we interviewed 46 researchers from diverse backgrounds. We investigated how they perceive their field, science communication, and the relation with religion. Results show that researchers actively participate in resolving the scientific debate, but delegate the resolution of controversies involving non-scientific institutions. Advocating for science is the foremost communication goal in all contexts. Career stage, research subject, religiosity, etc. influence the approach to controversies and communication.

## Introduction

The research on the origins of life (OoL) naturally touches on questions like "How did life come to be?" or "Where do we come from?". These issues have historically been the purview of philosophy and religious traditions. OoL research bring them to the natural sciences, spanning disciplines as diverse as physics, chemistry, biology, astronomy, and materials science. This diversity is both a blessing and a curse for the field. On the one hand, it fosters holistic and creative solutions; on the other, it deepens decade-old divides between approaches, focus points, and hypotheses originating from different fields [1].

We can picture the interaction between OoL research and society as a triangle (see Fig 1), connecting research, science communication, and religion. Despite a keen interest from the public, OoL can prove challenging for outreach. For instance, OoL research seldom focuses on applications, which would ease the communication with laypeople [2–5]. Moreover, the controversies in the field often revolve around relatively technical details. At the same time, OoL addresses existential, age-old questions of wide interest. Adding to the field's appeal, these questions may remain forever unsolved. At this junction, religion [6, 7] comes into play. Most religions and mythologies in the world include creation stories, which recount how the world, life, and human beings came to be [8].

The coexistence of scientific and religious narratives is a controversial topic in science communication. For some, these worlds can exist side-by-side, but for others, research and religion

—TRR 235 The funders had no role in study design, data collection and analysis, decision to publish, or preparation of the manuscript.

**Competing interests:** The authors have declared that no competing interests exist.

inevitably clash [9–11]. The very framing of the question drives the nature of the ensuing discussion [12], which can have wide societal ramifications, as the debate on creationism shows [12–16]. The scientific community has differentiated relationships with religion, much like the general public. Scientific communities are typically more secularized than the general population. For example 30–39% of Western-European researchers identify with "some religious affiliation" [17, 18]. 30–37% of scientists identify as non-believers or atheists, and an additional 10–28% as agnostic (with wide geographical differences) [17, 19]. In comparison, between 16% and 48% of Western-Europeans identify as non-affiliated (with wide differences between countries, median 24% [20]). In the EU overall, 27% of citizens identify as atheist, agnostic or non-believers [21]. However, like the general population, the scientific community presents a range of attitudes towards religious belief, spanning from peaceful separation [9] to acceptance, to full-on opposition [17, 18, 22].

With this work, we want to shed light on how OoL scientists navigate the relations between their research, its public communication, and religious beliefs. Concretely, we address three issues:

How do researchers see the controversy within the field of OoL research;

What approach do they prefer when communicating to the public;

How do they see–and wish to communicate–the controversy between science and religion in the context of OoL.

The sample population for this study are the researchers of the Munich-based Collaborative Research Center (CRC) 235 "Emergence of life". The CRC is a highly interdisciplinary research

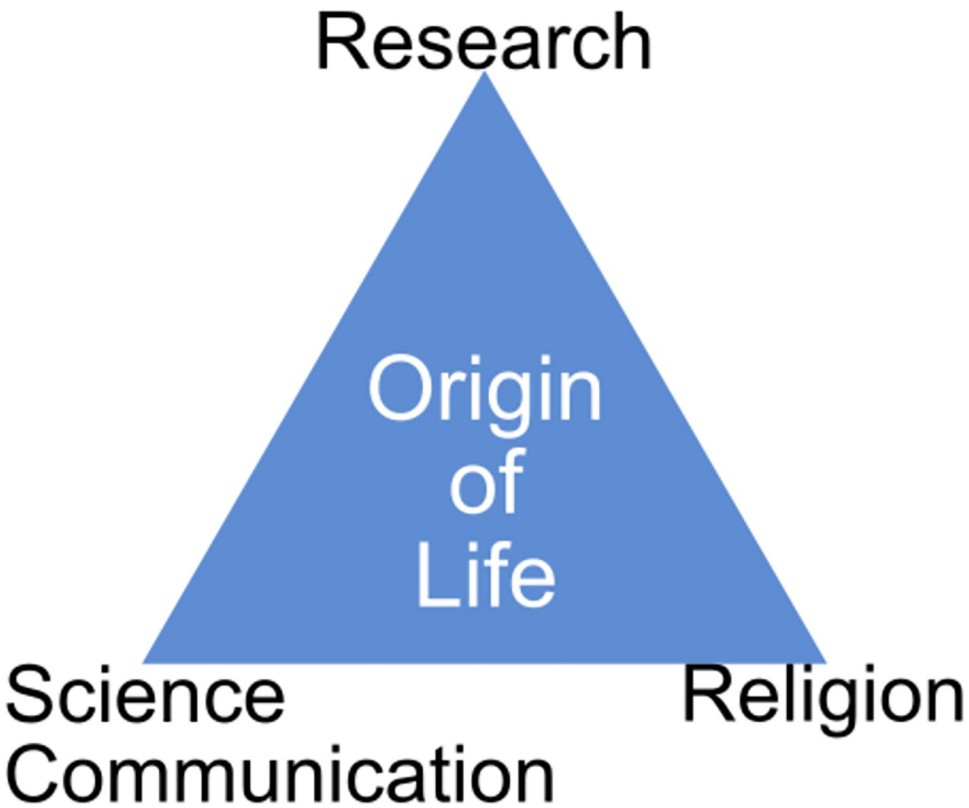

**Fig 1. The triangle of mutual social dependencies between OoL research, science communication, and religion.** These three aspects contribute to connect OoL to society and relate to each other. In this work we address how scientists approach these topics and their mutual dependencies.

network, bringing together scientists with diverse backgrounds to cooperatively work on different hypotheses and approaches to OoL research.

The results are presented in three parts: In the first we analyze the views that researchers have on the scientific controversy. In the second we analyze how scientists prefer to communicate the scientific controversy. In the last part, we focus on how scientists relate to religion and how that influences their communication patterns.

## Methods and sample

The data for this study originates from qualitative interviews conducted with members of the Collaborative Research Center (CRC) 235 "Emergence of life" between September 2019 and May 2020 as preparation for a planned museum exhibition about the emergence of life. Of 52 members of the CRC initially contacted (i.e., the entire network), six did not respond or declined the interview. Each workgroup/lab is represented in the survey by at least one member. Although the sample originates from only one research cluster, this is one of the largest on OoL research worldwide and is highly diverse in terms of professions, research backgrounds and hypothesis being followed. Therefore, while the study is not representative of the entire OoL field, it encompasses a significant slice of the population and offers comprehensive insights into the diverse mindset of OoL researchers. The final sample comprises 46 scientists at different career stages: 29 graduate students, 2 postdoctoral researchers, 2 non-professor faculty, and 13 professors. In terms of gender, 15 respondents identify as female, 31 as male. Most interviewees are German citizens, 13 have non-German citizenships (including one German-French binational) from a variety of countries in Europe, the Middle East, East Asia, and North America. 12 participants identified as atheist, 9 as agnostic, 14 as believing in some form of universal justice or spiritual force, 11 as affiliated to some denomination of Christianity: either as nominally affiliated, sporadically practicing, or regularly attending functions. The interviewees raised in religions different from Christianity no longer identify with any affiliation.

All 46 interviews were carried out in English, in which all participants are fluent (two native speakers). Most interviews (37) took place in person at the researchers' institutions, while nine were videocalls. All participants agreed to the recording of the interview and the anonymous collection of data. They were informed about the uses of the interview contents. Verbal consent was recorded in the interviews. The interview audios were transcribed and subsequently analyzed with the inductive method of qualitative analysis [23] to identify emerging topics. The categorization was carried out by one of the authors (KW) for all interviews, and by a second author (LK, independently) on a random sample (25%). Disagreements were negotiated until consensus was reached and applied uniformly to the remaining interviews.

Approval from an ethics committee was not sought for this study, as neither of the competent institutions (the Deutsches Museum and the TUM School of Social Sciences and Technology) had an appropriate council at the beginning of data collection. Furthermore, our study presented no risk of harm or injury for the participants, who were not subject to physical treatments, procedures or tests. The mental labor by the interviewees was limited to a conversation, in which they answered questions about their own perspective and experience on the field. No question was directed to the recall of negative memories. The extent to which to disclose such experiences, if at all, was entirely at the interviewee's discretion. Interviewees were also offered to opt out of any questions they would rather answer or not answer in full. Interviewees were all older than 18 and participated on an exclusively volunteer and non-compensated basis.

## Results

In the following sections we analyze the answers from the 46 researchers in our sample. First we focus on how they view the scientific controversies in the origins of life (OoL) field. Then on their approach to communicate these controversies. Finally we turn our attention to their attitude towards the relations between science and religion.

### Views about the scientific controversy

Most surveyed researchers perceive the scientific discussion about the origins of life (OoL) as conflictual. Table 1 presents the themes that emerged from the interviews regarding the controversy. Each theme was divided in classes of answers, shown in order of descending frequency. S1 Table presents more comprehensive and precise class definitions, including the specific rules that outline the boundaries of each class.

Only 9 of the 46 interviewees (20%) describe the scientific controversy as a competition between equally valid ideas, integral to the scientific sense-making process. For most the controversy is far from this Mertonian ideal [24]. According to many, the conflict stems from "dogmatic", "close-minded" attitudes, not rooted in "science". "Most of the discussion is really not an open and scientific discussion in the way it should be: people are in their little camps, and then they shoot to the other camp and tell them that they're doing stupid research." (19, Professor) Similar violations appear in as many as 33 interviews (69%), for 28 of which they are the most cited cause of conflict.

For some, the conflictual nature of the controversy traces back to single individuals. In Table 1, for example, the interviewee finds the people involved to be key for how the discussion unfolds ("it is always a history of person"). Conflict ensues only when people who do not "respect [each other]" are involved. More frequently (48% of cases), interviewees found the conflict to be driven by groups—sorted by hypothesis preference, discipline, or personal

**Table 1. Scientists' views about the scientific controversy.**

| Theme | Class description | Exemplary quote |
|---|---|---|
| Conflict actors | Groups or institutions collectively | There's definitely teams. (23) |
| | Single individuals | It's always a history of person. And if two persons [. . .] respect themselves, [the discussion] will be constructive, even though they have different opinions. (27) |
| Own role | Determine which side is right | I can, hopefully very objectively, decide if a process is possible or not. (35) |
| | Provide information | [W]hoever finally finds out which theory is the right one [. . .], I hope that I contributed enough information for the final answer. (28) |
| | Defend or advance a specific side | I hope that I would [. . .], excite people on the main road to try out the side road and widen their theoretical concept. (19) |
| | Actively try to defuse conflict | My role would be a mediator [. . .] I do think that I have input on how that discussion should be held. (9) |
| Cause of conflict | Active (willing) violation of rules or boundaries | You shouldn't be dogmatic about your research. And I think that's a lesson we have to learn. (8) |
| | Natural part of the process or of the actors involved | That's how we go from one point to more knowledge, right? To challenge the views, to challenge the beliefs, to challenge the established science. (21) |
| | There is no conflict | I went to several meetings where these questions were discussed [. . .] and there the discussion was very open. (15) |
| | Lack of information | [T]he chemists or the physicists, they think other about things. (37) |

allegiance. How the views about conflict actors correlate with those on the causes or the nature of the conflict is an interesting avenue of research. Our data, however, were insufficient to allow us to draw solid conclusions in that regard.

Almost half of the interviewed scientists (22) think that the problem is decreasing. Without any explicit prompt, they said that the most conflictual aspects are waning, thanks to the generational change and the increasing interdisciplinarity of the field: "So in some conferences . . . people are really shouting at each other. . . But sometimes I have the feeling . . . especially the younger people are much more open." (24, Professor).

To better summarize how the scientists themselves approach the controversy, we used their utterances in the "Own Role" category to build two conflict profiles. The answering profile actively participates in answering the scientific question—either by promoting a specific hypothesis, or as impartial judges: "This is for me something where I can, hopefully very objectively, decide if a process is possible or not." (35, Professor). The delegating profile, instead, delegates the decision to others, or brings the different sides to a discussion table: "I'm contributing valuable information to the topic. And whoever finally. . . gets the right answers, I hope that I contributed enough information." (28, Student) S2 Table reports the exact combinations of features that define each profile.

As shown in Fig 2, the predominant approach to the controversy changes with career stage, discipline, and hypothesis preference. Junior scientists (PhD students) frequently characterize themselves as "too low-rank" to contribute and so delegate the resolution. Meanwhile, senior scientists (postdoctoral researcher and higher) lean towards the answering profile. Tradition and gatekeeping could, in a similar vein, underlie the high fraction of answering approaches observed among chemists. This discipline, in fact, has the longest-standing controversies in the field (which contributes to more entrenched positions). As a result, chemistry is perceived as the dominant discipline in the field, sometimes excluding others: "I feel people are very focused on, 'If you're not a chemist, then you don't really know what you're doing.' Or, 'If you're not a chemist, then I don't believe in a lot of things you say anyway'." (23, Student).

## Communicating the scientific controversy

Table 2 shows how interviewees rated the importance of different aspects for a "good communication about the emergence of life" to a broad public, on a scale from 1 (negligible) to 6 (essential).

"Scientific basics" clearly have the highest rating. Because the survey was part of the preparations for a museum exhibition, the scientists might have been slightly biased in answering this question. Nevertheless, this singular focus on scientific basics also fits an emerging pattern. Indeed, other aspects are included, but subordinated to the goal of filling a gap in scientific knowledge. For example, historical theories serve as a good or captivating introduction to the modern view of facts. Some researchers also think about using their everyday work to engage the public and increase the acceptance of the results: "Everyday work of researchers: also super-fun. [. . .] It can just, you know, make [the topic] stick in people's minds" (25, Student). According to others, however, the daily grind of research may be distracting, boring, or even depressing: "Life as a researcher can be tough. And maybe people would feel a little bit down" (2, Student). As a result, this authenticity element scores quite low. A similar ambivalence underlies the low scores for potential applications: "In a sense, it's not important, but there will always be people who say, 'What is it good for?' [But when] you say, 'Well, if we create an artificial cell then you can also use it to make detergents' Maybe not" (41, Professor).

Table 3 presents the themes that emerged in interviews regarding the communication of Origin of Life (OoL) topics to a broad public. Each theme is divided in classes, shown in

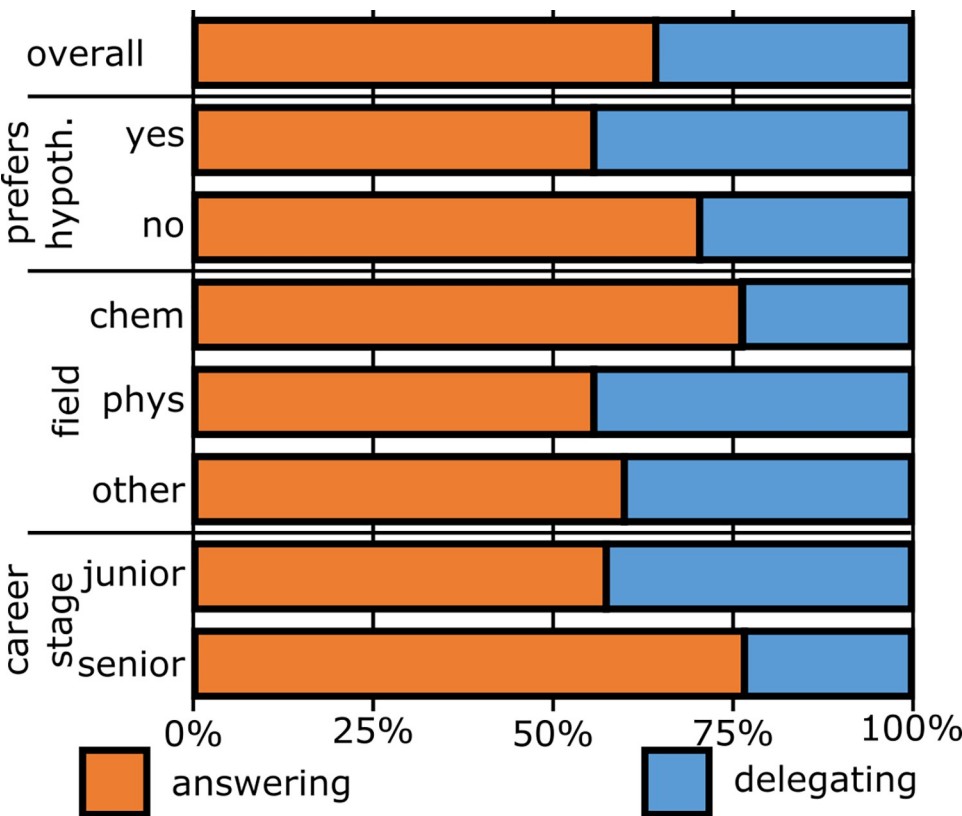

**Fig 2. Controversy profiles frequency.** Prevalence of the answering (orange) and delegating (blue) controversy profiles, when segmenting the sample by discipline, hypothesis preference, and career stage of the researchers (n = 45).

descending order of frequency. In this case, too, S1 Table collects more detailed definitions of each class.

We combined the statements of interviewees in the "Own role" and "Communication Model" theme to sort their communication approaches in one of three profiles (S3 Table for details). The teaching profile centers around the traditional deficit model. Teaching communicators unidirectionally transmit knowledge about scientific facts and the nature of science. "This notion of scientists only making statements that are based given some assumptions: it's

**Table 2. Importance scientists gave to different aspects of communication about origin of life.**

| Aspect | Mean Rating | Standard Deviation |
|---|---|---|
| **Scientific basics** | 5.17 | 1.19 |
| **Historical ("old") theories about life and its origins** | 4.20 | 1.35 |
| **Methods and processes of research** | 3.84 | 1.40 |
| **Relationship to religion and philosophy** | 3.61 | 1.54 |
| **Everyday work of a researcher** | 3.46 | 1.41 |
| **Relationship to potential applications (e.g. industrial chemistry, synthetic biology)** | 3.37 | 1.57 |

Mean and standard deviation of ratings given by scientists during the interviews. Ratings were expressed on a Likert scale from 1 (negligible) to 6 (essential). N = 46

**Table 3. Views about communication to a broad public.**

| Theme | Class description | Exemplary quote |
|---|---|---|
| Public attitude | Interested in topic or favors research | They're super interested. They really want to know. (20) |
| | Hostile towards research or topic | I know that a lot of people are opposed to [our work]. (10) |
| | Controversy or topic unknown to public | Over 95% of the population probably has no idea [the field] exists. (9) |
| | Neutral towards research | I think they would, in the first place, do not understand why we do [this research]. (28) |
| Target audience | Undifferentiated target audience | Everyone, everyone, really. So from kids to grandmas. (4) |
| | Single, uniform target segment | [P]eople who go to a museum and are interested in a little bit of science. (28) |
| | Differentiated targets | Everybody that goes to a presentation that is called "Origin of Life". And, yeah, also particularly children. (5) |
| Prior knowledge | Insufficient information or education | I think that many [. . .] are lacking basics [. . .], when they hear 'proteins', they think about going to the gym. (15) |
| | Public informed enough | If they're not curious to know, I think it probably is enough for them. (23) |
| | Public has wrong information | [A] certain part of community likes to write books. And then people read those books, and they believe this is the consensus in the field. (8) |
| Communication Model | Communication aims at filling knowledge gap | [M]ost of the people [. . .] do not know much about [OoL]. And it should be changed at some point, definitely. (18) |
| | Two-way information flow | Of course, they will care about [the research process]. Like they will ask you, "How did you sample it? How did you take your samples?" (6) |
| | Cooperative sense-making process | I would totally argue with [. . .] showing that there is [. . .] disagreement, and then let them take sides. And let's see what [. . .] comes from stakeholders on the street, [. . .] what could it be important aspects. (19) |
| Own role | Provide only interesting or understandable parts | It's a nice story you can tell. [. . .] It's easy to introduce the basic and to build [. . .] up. (1) |
| | Promote institution | It's kind of embarrassing to show to the public how researchers—which are thought to be the authority on knowledge—are fighting over if this molecule is prebiotic or not. (30) |
| | Present all facts as transparently as possible | We should always make clear, what are the scientific grounds? And where are we leaving the facts? [. . .] What are the conclusions? [. . .] What conclusions are [. . .] just speculative? (19) |
| | Patrol boundaries | To present this field to the public, I think it's important to stay within sciences that do use the scientific method and do not take just assumptions. (22) |

very important that people realize that" (16, Postdoc). The advocating profile aims at defending the institution of science, increasing its acceptance, or promoting the image of science and scientists. "It's not dangerous, we're not trying to clone or something" (15, Student). The discussing profile is less unidirectional. Discussing communicators report facts about the controversy for the sake of transparency or duty. "I think it will be nice to see the several approaches and where they're also heading to each other. . . And if you can make it some summary for every step of life that could be [based on] several approaches" (14, Student).

As Fig 3(A) shows, advocating is the most prevalent communication profile (21 interviewees), followed by teaching (10), while discussing is rather rare (3). 13 scientists did not clearly fit any of the profiles, thus were not taken into account in this analysis. Looking at the demographic breakdown, teaching is particularly prevalent among junior scientists, chemists, and those expressing a preference for any of the prevailing hypotheses for the origins of life (OoL). More senior scientists and physicists, instead, overwhelmingly tend towards advocating.

Fig 3(B) shows that different communication profiles are associated to different views about the public. Most interviewees—particularly in the advocating and discussing profiles—envision a public composed of one or more well-defined segments. The segments they mention reflect the typical audiences of informal scientific learning. One common example is visitors of science museums, families, and kids (clearly influenced by the exhibition activity

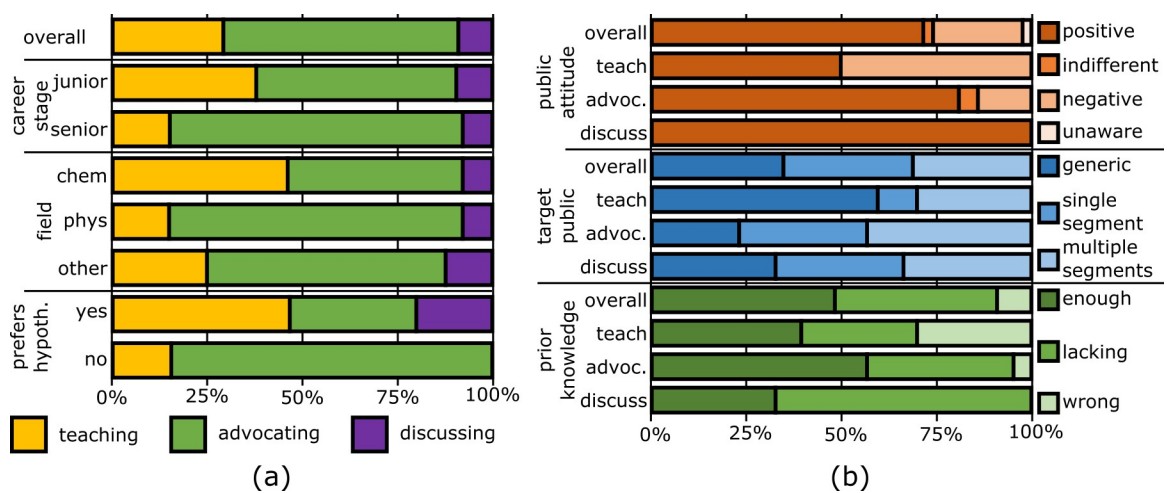

**Fig 3. Communication profiles prevalence and view of the public (n = 33).** (a) Prevalence of the teaching (yellow), advocating (green), and discussing (purple) profiles when segmenting the sample by career stage, field, and hypothesis preference of the researchers; (b) View of public attitude towards OoL (brown series), target public (blue series), and public's prior knowledge of OoL (green series) seen by all scientists (overall), as well as by the different communication profiles.

connected to the interviews). Another example are people that the researchers meet at their institutions' open days or at public lectures.

The interviewees fitting the teaching profile stand out from the average in their views of the public. For one, the majority imagines a generic, undifferentiated target public. Moreover, they more often expect the public to be wrongly informed about OoL research and to have a negative attitude towards it.

S1 Fig in the supplementary material also shows that different profiles prioritize different content in their communication. Teaching communicators, for example, focus more than average on history, scientific basics, and everyday life, while leaving methods and applications in the background. On the opposite end of the spectrum, advocating communicators prioritize relation to religion, methods, and applications more than their colleagues.

## Relating to religion and communicating the controversy

Origins of life (OoL), more than other research topics, shares a discussion space with non-scientific approaches, like philosophy and religion. This brings to the fore how scientists relate to belief.

We analyzed the interview contents using the same rules and classes deployed in the above sections (see Tables 1, 2 and S1–S3 Tables). Table 4 shows the prevalent answer classes in the themes that emerged discussing the scientists' views on dealing with the science-religion controversy and how to communicate about it. Most interviewees view science and religion as strictly independent and separated, although not intrinsically in conflict. Those that see conflict take the side of science: "There's a lot of conflict, but there doesn't have to be. If there were to be no conflict, it would be religion who would have to accommodate" (9, Student). Almost one third of the interviewees (28%) said—unprompted—that they experienced or imagine the conflict being more pronounced in the United States or countries with a strong Islamic influence than it is in the Germany and Western Europe.

Fig 4(A) shows the distribution of religious self-identification among scientists in our survey. The sample splits roughly equally between atheist, agnostic, spiritual (who believe in some

**Table 4. Scientists' views about the science-religion relationship and its communication.**

| Theme | Class description | Exemplary quote |
|---|---|---|
| Conflict actors | Groups or institutions collectively | Religion's really not the promoter of science. [...], there will be conflicts, definitely. (34) |
| | Single individuals | I think in some people beliefs are very contradicting, but I think [...] they can complement each other. (17) |
| Own role in conflict | Advance specific side | If we do our jobs well we will be able to prove one day that the origin of life didn't have to be mystical, it didn't have to be created by God. (9) |
| | Provide information | I think we should engage these people [...] and say, "[...] Earth is four and a half billion years old. And after 3.8 billion years there was life [...]" And maybe they'll just walk away and not believe you but at least they heard it. (33) |
| | Actively try to defuse conflict | Why not to make connection with what was traditionally explained [...] and what is the scientific explanation or interpretation? [A]lso science is an interpretation of reality. (40) |
| | Determine which side is right | [W]e can just generate facts and if questions come, then we have to think about it. (27) |
| Cause of conflict | Lack of information | If you think that God created the Earth in seven days [...] that's not gonna work. (11) |
| | There is no conflict | [S]cience and the beauty of mathematics, [...], of how the world works, and how beautiful and how logical everything is, this is for me the best proof that there's some God. (5) |
| | Active (willing) violation of rules or boundaries | There's a lot of conflict, but there doesn't have to be. [...] If there were to be no conflict, it would be religion who would have to accommodate. (9) |
| Own role in communication | Patrols boundaries between institutions | I think already showing that the origin of life research is not just research to disprove that God exists. (12) |
| | Presents all facts transparently | But in principle, historically spoken, it's... it's... of course it's important and one should consider and keep in mind. (32) |
| | Promotes an institution | [N]ot so much as an equivalent theory, because [...] one is popular belief, and the others are being tested through the scientific method. (22) |

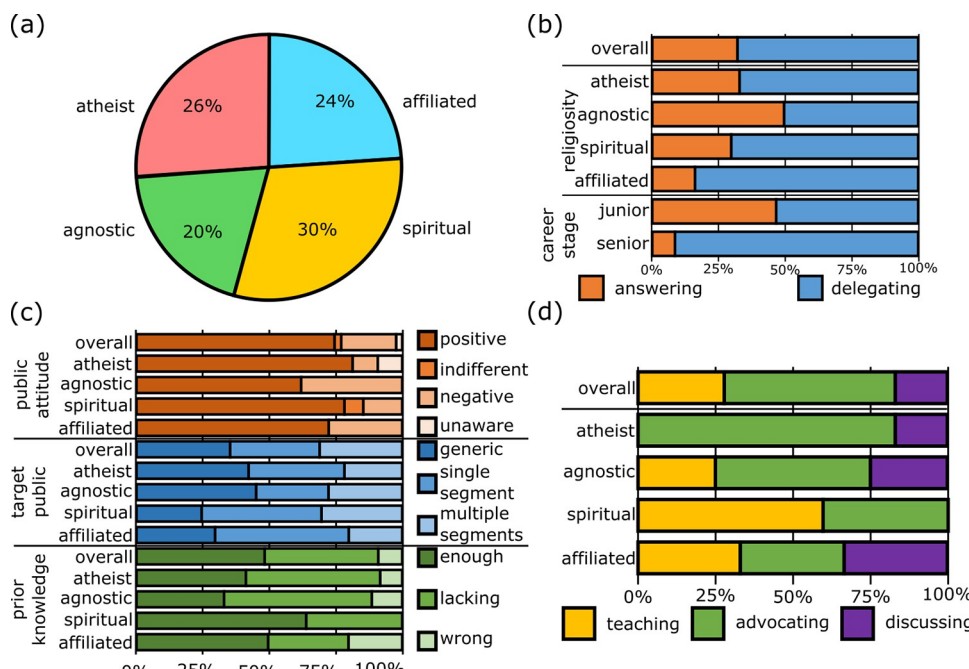

**Fig 4. Religion and science in origins of life.** (a) Religious composition of the sample, roughly equally divided between atheists, agnostics, spiritually inclined and religiously affiliated to some degree (N = 46). (b) Prevalence of answering (orange) and delegating (blue) profiles in the science-religion controversy (n = 28). (c) How different religious segments see: the public attitude towards OoL (brown series), the target public (blue series), and the public's prior knowledge of OoL (green series) (N = 46). (d) Prevalence of the teaching (yellow), advocating (green), and discussing (purple) profile by religiosity (n = 18).

form of universal justice or spiritual force), and religiously affiliated (ranging from nominally affiliated to sporadically practicing to regularly attending functions).

To analyze the scientists' attitude towards the science-religion controversy, we followed the same method as for the scientific controversy, namely creating profiles. Some interviewees fit the answering profile and actively participate in resolving the controversy. Others fit the delegating profile: they foster dialogue and delegate the conflict resolution. The delegating profile is much more prevalent in the controversy with religion (64%) than in the scientific controversy (32%). Furthermore, this profile becomes even more frequent among senior scientists. This trend is diametrically opposed to what we observed in the scientific controversy (see Fig 2).

Segmenting the sample by religiosity, we see that scientists identifying with a religious affiliation are less likely to fit the answering profile. However, the delegating profile is also prevalent among self-identified atheists. In the context of the scientific controversy, increased religiosity corresponds to a more frequent answering profile (see S2 Fig).

We also considered how interviewees communicate the controversy arising from the different value systems of science and religion. 33 of them (72%) said their communication should primarily trace boundaries between the domains, either by clarifying differences between the approaches, or pointing out limitations—sometimes with the explicit goal of preventing or defusing the conflict: "[We should show] that the origin of life research is not just research to disprove that God exists" (12, Professor) "We should always make clear: what are the scientific grounds? And where are we leaving the facts? Well, what are the conclusions? What are the possible conclusions? And what conclusions are [. . .] just speculative?" (19, Professor) Thus the most widespread approach aligns with S.J. Gould's "Non-Overlapping Magisteria" argument [9]. There are, however, notable exceptions. Some interviewees, for example, wanted to explicitly connect science and religion as different expressions of the same sense-making process. "They're connected like everything that is part of our culture. Why not to make connection with what was the interpretation that was given in that field? Also science is an interpretation of reality." (40, Postdoc).

As Fig 4(C) shows, different religiosities were associated to different communication approaches in our interviews. Religiously affiliated and spiritually inclined scientists, for example, are least likely to envision a generic, undifferentiated audience. They also see the public as informed "enough" and with a positive attitude towards OoL research. Self-identified agnostics, instead, have the most pessimistic view of the public: they are the least likely to perceive a positive public attitude or to see the public as informed "enough" about OoL. Agnostics were also the most likely to target an undifferentiated public with their communication.

Finally, we summarized the scientists' communication approach in three profiles (Fig 4(D)): a deficit-model-based teaching profile, an advocating profile focused on promoting and defending science (none of the interviewees presented themselves as defending religion over science), and a dialogue-oriented discussing profile. Though the small sample limits the scope of conclusions, we can see atheists taking a predominantly advocating profile, whereas spiritually inclined researchers tend to the teaching profile.

## Discussion

We interviewed 46 scientists from the Collaborative Research Center 235 "Emergence of Life" (CRC) to explore: (i) how they view the scientific controversies in the origins of life (OoL) field; (ii) their approach to communicate them; and (iii) their attitude towards the relations between science and religion. Although the sample encompasses members of only one research cluster, it is diverse enough to offer a snapshot of the attitudes in the field (see Methods and Sample). The most frequent approach to the OoL controversy is to actively participate

in its resolution, particularly among senior scientists. The controversy with religion, meanwhile, presents the exact opposite scenario: a more cooperative and delegating approach, especially in senior scientists. The communication of both controversies aims most often to advocate for science. The specific communication profile each individual fits is also linked to their view about the target public.

Laherto *et al*. already observed that, when approaching the public, scientists sometimes prioritize to communicate other aspects of the field, rather than what they find important in research work [25]. Here we also see that communication and controversy profile do not appear linked. For example, the approach to controversy among chemists resembles that of senior scientists, while their communication profile is closer to that of junior scientists.

The more passive, delegating approach of junior scientists is not surprising, as they are professionally subordinate. We can therefore expect current graduate students to become more answering as they gain experience, confidence, and status. However, there is also a widespread sense (across different career stages) that the generational shift is leading the discourse towards a more constructive dialogue and away from conflict and dogmatic positions.

Our results confirm that state-of-the art science communication cannot be demanded from scientists "out of the box". Their direct involvement is crucial for an engaging communication. However, untrained researchers have a natural tendency towards the deficit model. Experience and training in public communication counteract this trend. Our results also suggest that communication experience alone moves the needle towards advocating for science rather than to dialogic models. Therefore, specific communication training is also necessary before expecting scientists to adhere to those modern models.

The scientists surveyed here have very different communication priorities than, for example, the researchers in nanoscience and -technology (NST) surveyed in [25]. Prominently, scientific basics are the top priority for OoL researchers, but next to the bottom for nanoscientists. The scientific process was also frequently mentioned in our interviews, but had much lower priority in NST. Meanwhile, methods were often classified as too complex in our survey, and thus were given much lower priority than in surveys of nanoscientists. Finally, applications offer a curious point of comparison. Both NST and OoL researchers viewed applications as marginal to their daily work. Nevertheless, nanoscientists put applications, products, and usefulness at the top of their communication priority. OoL researchers, instead, gave applications the lowest priority rating, despite the close links of their research to potentially hot topics, such as synthetic biology.

Our sample includes a wide spectrum of religious self-identification, almost evenly split between atheists (26%), agnostics (20%), spiritual (30%), and religiously affiliated (24%). This distribution roughly aligns with comparable surveys in Europe and North America [17–19], in which 30–37% of the scientists identify as atheist, 10–28% as agnostic, and 30–39% identifies with "some religious affiliation" [17, 18]. Also compared religiosity in scientists and in the general population of the same country. Consistently with those findings, we also see non-affiliation being much more prevalent in our sample than in the general (Western-) European population. In polls from 2018 and 2019, less than 30% of Europeans identified as atheist, agnostic, or generally non-believer [20, 21] whereas atheists and agnostics alone make up 46% of our sample.

Most surveyed scientists subscribe to the idea that science and religion must be kept separate as much as possible. To that end, they engage in boundary work, similarly to what was observed in previous studies [17, 22], for example so-called conciliatory boundary work [22], dividing "good" religion (secular, accommodating, and compatible with science) from "bad" religion (dogmatic, incompatible with science). "If you think that God created the Earth in seven days that's not gonna work. But if you think that maybe God is more a force of nature or something, then yes." (11, Student).

The interviewees' religiosity seems to be associated with their communication approach. Religiously affiliated and spiritually inclined scientists, for example, expressed the most positive view of the public. Self-identified agnostics, instead, were the most pessimistic, more than atheists. The data gives no conclusive evidence on what underlies this difference. One could suspect that atheists are more actively engaged in the controversy, hence their profile resembles that of experienced communicators. Conversely, some of the people who view the debate negatively might identify as agnostic to avoid engaging with it.

OoL is an exciting, lively field with a strong potential to involve a diverse public within academia and outside of it. Our study portrays a considerable diversity of views, goals, and approaches. As a tool to create outreach, the study offers a snapshot to convey an authentic picture of OoL research, as well as the professional and personal views of the scientists behind it.

The following are available online at https://doi.org/10.6084/m9.figshare.14703060:

## Supporting information

**S1 Fig. Communication aspects priority for communication profiles.** How different communication profiles prioritize aspects of communication.
(PDF)

**S2 Fig. Additional demographics for controversy and communication profiles.** Controversy and communication profiles related to the scientific controversy and the controversy with religion, segmented by field of work, hypothesis preference, seniority, religiosity, number of countries researchers lived in.
(PDF)

**S1 Table. Classification rules.** Classification table with rules that defined each of the classes and profiles.
(PDF)

**S2 Table. Definitions of controversy profiles.** Composition of each controversy profile in terms of most frequent answer classes.
(PDF)

**S3 Table. Definitions of communication profiles.** Composition of each communication profile in terms of most frequent answer classes.
(PDF)

**S1 File. Minimal dataset.** Anonymized minimal dataset to replicate all figures and conclusions.
(XLSX)

## Acknowledgments

We wish to thank the members of the CRC 235 Emergence of Life for their kind and generous cooperation during the interview process and Zara Gough for language help.

## Author Contributions

**Conceptualization:** Karl Wienand, Lorenz Kampschulte.

**Formal analysis:** Karl Wienand, Lorenz Kampschulte.

**Funding acquisition:** Wolfgang M. Heckl.

**Investigation:** Karl Wienand, Lorenz Kampschulte.

**Methodology:** Lorenz Kampschulte.

**Project administration:** Wolfgang M. Heckl.

**Supervision:** Wolfgang M. Heckl.

**Writing – original draft:** Karl Wienand, Lorenz Kampschulte.

**Writing – review & editing:** Karl Wienand, Lorenz Kampschulte, Wolfgang M. Heckl.

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
