## [Decision Letter · Decision Letter 0]

12 Sep 2022

PONE-D-22-02824Creating a foundation for Origin of Life outreach: How scientists relate to their field, the public, and religionPLOS ONE

Dear Dr. Wienand,

Thank you for submitting your manuscript to PLOS ONE. After careful consideration, we feel that it has merit but does not fully meet PLOS ONE’s publication criteria as it currently stands. Therefore, we invite you to submit a revised version of the manuscript that addresses the points raised during the review process.

Please note that we have only been able to secure a single reviewer to assess your manuscript. We are issuing a decision on your manuscript at this point to prevent further delays in the evaluation of your manuscript. Please be aware that the editor who handles your revised manuscript might find it necessary to invite additional reviewers to assess this work once the revised manuscript is submitted. However, we will aim to proceed on the basis of this single review if possible.  The reviewer has made a number of suggestion for improvement (see comments below). 

Could you please carefully revise the manuscript to address all comments raised?==============================

We look forward to receiving your revised manuscript.

Kind regards,

Steve Zimmerman, PhD

Associate Editor, PLOS ONE

“Karl Wienand was funded by the Deutsche Forschungsgemeinschaft (DFG, German Research Foundation) with Project-ID 364653263—TRR 235”

“We wish to thank the members of the CRC 235 Emergence of Life for their kind and generous cooperation during the interview process and Zara Gough for language help. KW is grateful to the Deutsche Forschungsgemeinschaft (DFG, German Research Foundation)—Project-ID 364653263—TRR 235 for funding.”

“Karl Wienand was funded by the Deutsche Forschungsgemeinschaft (DFG, German Research Foundation) with Project-ID 364653263—TRR 235”

6. We note that you have indicated that data from this study are available upon request. PLOS only allows data to be available upon request if there are legal or ethical restrictions on sharing data publicly. For more information on unacceptable data access restrictions, please see http://journals.plos.org/plosone/s/data-availability#loc-unacceptable-data-access-restrictions.

Reviewers' comments:

Reviewer's Responses to Questions

**Comments to the Author**

1. Is the manuscript technically sound, and do the data support the conclusions?

Reviewer #1: Yes

2. Has the statistical analysis been performed appropriately and rigorously? 

Reviewer #1: I Don't Know

3. Have the authors made all data underlying the findings in their manuscript fully available?

Reviewer #1: No

4. Is the manuscript presented in an intelligible fashion and written in standard English?

Reviewer #1: Yes

5. Review Comments to the Author

Reviewer #1: This article tackles an interesting subject. A number of revisions could improve the final version. Some are small, eg in line 47 the authors state that the majority of research on creationism has taken place in the US but they could also easily reference recent contributions on Europe. Similarly in lines 49-56 statistics are compared regarding scientists in Western Europe and the general German population- it would be better if the comparison groups were the same' ie German scientists w the German general population or Western European scientists with the general Western European population. Additionally, in lines 91-94 the religious identity of the scientists was described and it was not clear regarding those scientists of non-Christian backgrounds- there were no Muslim, Hindu or Jewish scientists? This needs some further elaboration. On the other hand it seemed that the material in lines 104-116 could be switched to a note and the sentences in lines 310-313 could be moved to an earlier section. Some of the quotations brought from the interviews were not always clear, eg Table 3, neutral towards research: "I think they would, in the first place, do not understand why we do it." There were also some occasional lines in the text that should be adjusted (e.g. line 369). The main substantive question I have is regarding the formulation of the tables. More discussion of the categories would be helpful to explain the choices made by the researchers and to describe further the contours of the typology they found (eg the examples brought for single actors vs groups were more focused on the constructive nature of differing views where as those who saw camps seemed to describe this in competitive terms-- is this significant, coincidental etc?) I look forward to seeing this further explication and expanded discussion as I am very interested in this research.

6. PLOS authors have the option to publish the peer review history of their article (what does this mean?). If published, this will include your full peer review and any attached files.

Reviewer #1: No

---

## [Author Response · Author response to Decision Letter 0]

24 Oct 2022

Dear Editor,

Thank you very much once again for considering my paper for publication in PLOS ONE.

Based on the Editor and Reviewer’s suggestions and comments, we were able to extensively edit the Manuscript. We clarified, improved, and extended the discussions as much as possible. Moreover we substantially expanded the supplementary materials to address the concerns of more specialized and/or interested readers.

Below our answers to each of the Reviewer’s comments.

Sincerely,

Karl Wienand

• In line 47 the authors state that the majority of research on creationism has taken place in the US but they could also easily reference recent contributions on Europe.

We thank the Reviewer for pointing this out. Indeed, limiting the discussion of creationism debates to the US environment could be reductive. We expanded the references (particularly Refs 14—16) to include the European context, as well as debates around non-Christian creationism.

• In lines 49-56 statistics are compared regarding scientists in Western Europe and the general German population- it would be better if the comparison groups were the same' ie German scientists w the German general population or Western European scientists with the general Western European population.

We thank the reviewer for the comment. Comparing more uniform data provides a better picture of secularization among scientists. We researched different sources (i.e., Refs 20, 21) to compare our sample with Western European and EU general population.

• In lines 91-94 the religious identity of the scientists was described and it was not clear regarding those scientists of non-Christian backgrounds- there were no Muslim, Hindu or Jewish scientists? This needs some further elaboration. 

No, the sample happened to not include any scientists with Hindu or Jewish background. One was raised Muslim but no longer identifies as affiliated. We amended the statement for clarity, but left the part still intentionally vague to protect the participant’s anonymity.

• The material in lines 104-116 could be switched to a note

These considerations, though admittedly not necessary to replicate the study, had to be included in the Methods section following a request from PLOS ONE.

• The sentences in lines 310-313 could be moved to an earlier section.

We thank the reviewer for the suggestion. We added a similar text to the beginning of the Results section. However, we also left the original lines in place, as a quick recap for readers who may have skipped directly to the Discussion section.

• Some quotations brought from the interviews were not always clear

We thank the reviewer for the suggestion. In preparing the manuscript we had tried to leave the quotes as much as possible in their original form. As the reviewer points out, however, this caused problems, as it was sometimes unclear what the interviewees were referring to with “it”, “they”, etc. We edited quotes throughout the manuscript for clarity. All alterations to the original transcripts are denoted in square brackets. For example, the cited quote in Table 3 now reads: “I think they would, in the first place, do not understand why we do [this research].”

• Some occasional lines in the text that should be adjusted (e.g. line 369)

We thank the reviewer for the comment. We polished the entire text, of course starting from that line.

• More discussion of the categories would be helpful to explain the choices made by the researchers and to describe further the contours of the typology they found (eg the examples brought for single actors vs groups were more focused on the constructive nature of differing views where as those who saw camps seemed to describe this in competitive terms-- is this significant, coincidental etc?)

We thank the reviewer for the keen interest in the topic and the thoughtful remark. The example quote for “Single actors” was chosen because it stresses how the conflictual nature depends, according to this interviewee, only on the people involved (“it's always a history of person”). We expanded the manuscript text to clarify this example as well as the overall category.

We also agree that interested readers should have access to a more detailed treatment of our classification process and rules. During the analysis we worked with a more comprehensive table than the one shown in the manuscript. This complete classification table included specific a priori rules that determined to which class a sentence would be assigned. While we still believe the summarized table in the main manuscript is more accessible for the broad readership of PLOS ONE, we agree that it may leave the boundaries too blurry for the interested reader. Therefore we added to the supplementary material the full rule table (S1 Table) that was used to analyze the interview content, as well as the rules delineating each the controversy and communication profiles (Tables S2 and S3, respectively).

---

## [Decision Letter · Decision Letter 1]

13 Feb 2023

Creating a foundation for Origin of Life outreach: How scientists relate to their field, the public, and religion

PONE-D-22-02824R1

Dear Karl Wienand

We’re pleased to inform you that your manuscript has been judged scientifically suitable for publication and will be formally accepted for publication once it meets all outstanding technical requirements.

Kind regards,

Margaret Williams, Ph.D

Academic Editor

PLOS ONE

Additional Editor Comments (optional):

Reviewers' comments:

Reviewer's Responses to Questions

**Comments to the Author**

1. If the authors have adequately addressed your comments raised in a previous round of review and you feel that this manuscript is now acceptable for publication, you may indicate that here to bypass the “Comments to the Author” section, enter your conflict of interest statement in the “Confidential to Editor” section, and submit your "Accept" recommendation.

Reviewer #1: (No Response)

2. Is the manuscript technically sound, and do the data support the conclusions?

Reviewer #1: (No Response)

3. Has the statistical analysis been performed appropriately and rigorously? 

Reviewer #1: (No Response)

4. Have the authors made all data underlying the findings in their manuscript fully available?

Reviewer #1: (No Response)

5. Is the manuscript presented in an intelligible fashion and written in standard English?

Reviewer #1: (No Response)

6. Review Comments to the Author

Reviewer #1: Comments-

Specifics-

1- I sent emails regarding my comments on the abstract provided and a different version I found in the online supplementary file, but did not get a response (edits I had originally suggested on the abstract are below as are copies of the emails I sent).

2- The phrase “blessing and curse” is a bit strong and line 25 can be adjusted.

3- In line 30 the word “scientific” should be added before research to clarify.

4- Lines 37-38 could be expanded. These issues touch on people’s identity and self-understandings, but perhaps that will be in future publications.

5- The statistics offered in lines 51-57 are still not entirely parallel and clear, e.g. the percentage in the scientific community who identify with some religious affiliation is given, but not the parallel percentage in the general population.

6- Line 63— perhaps “controversies” instead of “the controversy”?

7- Lines 73-74 refers to “the scientific controversy” again—it seems this controversy should be explained/detailed. The phrase comes up later again as well e.g. line 156 etc- the controversy or even main controversies in the field call out for a line or two of explanation for the reader.

8- Line 116-17 need slight adjustment—people could opt out of questions they would rather not answer or not answer in full.

9- Line 139-40 Perhaps: Similar accusations (instead of violations)? Second half of sentence not entirely clear, perhaps: dogmatic attitudes are the most cited source of conflict?

10- 171-72- Which “discipline”?

11- 183 no “a” before “good”

12- Line 359- out of the box may not be the clearest phrase to use here

13- 383 consistent

I did not do a fully copy editing of the document, but rather pointed out some issues that jumped out at me. Additionally while obviously quotations from informants need to be reported accurately, some of the quotes could use further adjustment for clarity, e.g.: I think already showing that the origin of life research is not just research to disprove that God exists. (12) It's kind of embarrassing to show to the public how researchers—which are thought to be the authority on knowledge—are fighting over if this molecule is prebiotic or not. (30) I think they would, in the first place, do not understand why we do [this research]. (28)

General comment and question—

1- As I wrote in the initial review, I believe this subject of the interface of scientific research and science communication in the development of museum exhibits to be extremely interesting and important. I think that the authors have brought forth valuable new information. I also think that the article would benefit from having a second reviewer. I could offer a potential recommendation if needed.

2- The authors say that all of the data is available, but this does not include recordings or transcriptions of interviews—what is required to be available for a qualitative study?

Editing I had previous done of abstract:

11 Origins of life (Ool) research is particularly challenging to communicate because of the

12 tension between its many disciplines and its nearness (find better word, connection?) to traditionally philosophical or

13 religious questions. To authentically represent scientists’ perspectives in the development of a museum

14 exhibition on Ool, we interviewed 46 researchers from diverse backgrounds (their backgrounds were diverse in some ways but not others correct? Perhaps this needs further thought?). We investigated

15 how they perceive their field (what do you mean? In terms of…?), (the nature of?) science communication, and the relation (of what?) with religion.

16 Results show that researchers actively participate in resolving the scientific debates, but

17 delegate the resolution of controversies involving non-scientific institutions (to whom?). Advocating

18 for science is the foremost communication goal in all contexts. Career stage, research

19 subject, religiosity, etc. (etc. is not great to use- suggest to adjust accordingly) influence the approach (adopted) to (relate to?) controversies and communication.

Emails I sent:

Question about article I am reviewing PONE-D-22-02824R1

Inbox

Mon, Nov 7, 4:12 PM (10 days ago)

to PLOS

Dear Editor,

I am currently reviewing an article, and I have noticed something that I would like to discuss with you. When I went to the website where supplementary information for the manuscript is stored, I found a better version of the abstract of the paper than that which I received. In fact I had begun to suggest a number of edits to the abstract as it was sent to me, but in the version I then found online the issues I had pointed to are already resolved. This makes me wonder if I have the correct, most updated, version of the article for review. I will paste the abstracts below, and will await advice from you. The rest of the paper could also use additional language editing, and of course would be a shame if I am wasting time on an old version. There are substantive comments to make, but the linguistic issues have been distracting.

Thank you,

The abstract in the manuscript sent to me:

Abstract

11 Origins of life research is particularly challenging to communicate because of the

12 tension between its many disciplines and its nearness to traditionally philosophical or

13 religious questions. To authentically represent scientists’ perspective in a museum

14 exhibition, we interviewed 46 researchers from diverse backgrounds. We investigated

15 how they perceive their field, science communication, and the relation with religion.

16 Results show that researchers actively participate in resolving the scientific debate, but

17 delegate the resolution of controversies involving non-scientific institutions. Advocating

18 for science is the foremost communication goal in all contexts. Career stage, research

19 subject, religiosity, etc. influence the approach to controversies and communication.

The improved abstract I found online (https://figshare.com/articles/figure/Supplementary_information_to_Creating_a_foundation_for_Origin_of_Life_outreach_How_scientists_relate_to_their_field_the_public_and_religion_/14703060/3)

Origins of life (OoL) is a multi-disciplinary field at the cutting edge of research. At the same time, the field touches on traditionally philosophical and religious questions, like “What is life?” and “Where do we come from?” This tension makes it particularly challenging for science communi-cation. To better understand OoL researchers and to authentically incorporate their views in a planned exhibition project, this study analyses the perspectives scientist have on their filed, on science communication, and the relation of science and religion. A total of 46 researchers from diverse disciplines and backgrounds were interviewed. While they actively participate and re-solve the scientific debate within OoL, they keep religion separate and delegate the resolution of conflicts with it. The communication of both controversies aims most often to advocate for sci-ence. Conflict and communication profiles are linked to a variety of factors such as career stage, research subject, and religiosity.

Fri, Nov 11, 2:34 PM (6 days ago)

to PLOS

Hello-

I wrote earlier in the week with questions regarding discrepancies I found between versions of the abstract of the article in the on line supplemental folder and the one in the paper I downloaded. If in any case I should just review the article as is, I will work on it next week. If there is a response to the question I sent I would be glad to hear.

7. PLOS authors have the option to publish the peer review history of their article (what does this mean?). If published, this will include your full peer review and any attached files.

Reviewer #1: No

---

## [Editor Report · Acceptance letter]

15 Feb 2023

PONE-D-22-02824R1 

Creating a foundation for Origin of Life outreach: How scientists relate to their field, the public, and religion 

Dear Dr. Wienand:

I'm pleased to inform you that your manuscript has been deemed suitable for publication in PLOS ONE. Congratulations! Your manuscript is now with our production department. 

Kind regards, 

on behalf of

Professor Margaret Williams 

Academic Editor

PLOS ONE